**Multi-scheme chemical ionization inlet (MION) for fast switching of reagent ion chemistry in atmospheric pressure chemical ionization mass spectrometry (CIMS) applications**

Matti P. Rissanen[1, 2], Jyri Mikkilä[3], Siddharth Iyer[2,4], Jani Hakala[3]

[1]Department of Physics and Institute for Atmospheric and Earth System Research, University of Helsinki, Helsinki, Finland

[2]Aerosol Physics Laboratory, Physics Unit, Faculty of Engineering and Natural Sciences, Tampere University, Tampere, Finland

[3]Karsa Ltd., A. I. Virtasen aukio 1, 00560 Helsinki, Finland

[4]Department of Chemistry and Institute for Atmospheric and Earth System Research, University of Helsinki, Helsinki, Finland

*Correspondence to*: Matti P. Rissanen (matti.rissanen@tuni.fi), Jani Hakala (jani.hakala@karsa.fi)

**Abstract.** A novel chemical ionization inlet (*Multi-scheme chemical IONization inlet*, MION, Karsa Ltd, Helsinki, Finland) capable of fast switching between multiple reagent ion schemes is presented and its performance is demonstrated by measuring several known oxidation products from much studied cyclohexene and α-pinene ozonolysis systems, by applying consecutive bromide ($Br^-$) and nitrate ($NO_3^-$) chemical ionization. Experiments were performed in flow tube reactors under atmospheric pressure and room temperature (22°C) utilizing *atmospheric pressure interface time-of-flight mass spectrometer* (APi-ToF-MS, Tofwerk Ltd, Thun, Switzerland) as the detector. The application of complementary ion modes in probing the same steady-state reaction mixture enabled a far more complete picture of the detailed autoxidation process; the $HO_2$ radical and the least oxidized reaction products were retrieved with $Br^-$ ionization, whereas the highest oxidized reaction products were detected in the $NO_3^-$ mode, directly informing on the first steps and on the ultimate end-point of oxidation, respectively. While chemical ionization inlets with multiple reagent ion capabilities have been reported previously, an application in which the charging of the sample occurs at atmospheric pressure with practically no sample pretreatment, and with the potential to switch the reagent ion scheme within a second time-scale, has not been introduced previously. Also, the ability of bromide ionization to detect *highly-oxygenated organic molecules* (HOM) from atmospheric autoxidation reactions has not been demonstrated prior to this investigation.

**Keywords:** Mass Spectrometry, Chemical ionization, CIMS, CI-APi-ToF, HOM, Peroxy radicals, $HO_2$ and $RO_2$, Autoxidation, Atmospheric Acids

# 1 Introduction

*Chemical ionization mass spectrometry* (CIMS) is a versatile analysis technique that enables detection of gas-phase molecular constituents at atmospheric pressure and at concentrations as low as $10^5$ $cm^{-3}$ (Munson and Field, 1966; Munson, 1977; Eisele and Tanner, 1993; Huey, 2007; de Gouw and Warneke, 2007; Mauldin et al., 2012; Sipilä et al., 2015; Hyttinen et al., 2018; Laskin et al., 2018). With the right selection of reagent ions that either form adducts with the analytes or transfer their electric charge (*e.g.,* with an electron or a proton transfer), CIMS can offer a soft, selective, and extremely sensitive online detection for virtually any gas-phase chemical compounds. In recent years, various CIMS methods have revolutionized the ways we understand atmospheric *volatile organic compound* (VOC) oxidation processes (see *e.g.,* Ehn et al., 2014; Rissanen et al., 2014; Jokinen et al., 2015; Kirkby et al., 2016; Lee et al., 2016; Breitenlechner et al., 2017), especially the routes leading to oxidative molecular growth [*i.e.,* fast autoxidation (Crounse et al., 2013; Berndt et al., 2015; Rissanen et al., 2015] and slow aging (Donahue et al., 2006; Hallquist et al., 2009)] and subsequent *secondary organic aerosol* (SOA) formation.

In principle, a mass spectrometer is a universal detector with applicability mainly controlled by two factors: (i) volatility, and (ii) ionizability of the analyte (Baeza-Romero et al., 2011; McLafferty, 2011). In practice, the sampled material must be volatilized into the gas-phase and then ionized by a suitable method. In a gas-phase atmospheric application, the problem reduces to the latter, as the analytes are inherently aloft in the surrounding gas media. Unfortunately, the most universal ionization methods, by definition, lack the selectivity needed for separating compounds from complex gas mixtures and information from multiple complementary techniques are generally required to enable chemical speciation (*e.g.,* isomer separation), with ambiguity nevertheless quickly increasing with the complexity of the target molecule. The utilization of multiple ionization schemes in CIMS has the potential for detailed chemical speciation of the target compounds by exploiting chemical selectivity, *e.g.,* characterizing amines by ethanol CIMS (Yu and Lee, 2012), peroxy acids by $I*H_2O^-$ (Iyer et al., 2017), and hydroperoxides by $CF_3O^-$ (Crounse et al., 2006). However, CIMS instruments and their ion sources are expensive and bulky, and consequently, most research laboratories have access to a precious few. Ideally, a versatile, easily deployable chemical ionization source capable of rapid switching between multiple reagent ions with varying chemical selectivity would be indispensable in tackling the complexity encountered in various gas-phase environments.

CIMS inlets with multiple reagent ion capabilities have been reported previously (*e.g.,* Jordan et al., 2009; Brophy and Framer, 2015), perhaps most commonly in applications concerning *proton transfer reaction mass spectrometry* (PTRMS). In addition to the usual $H_3O^+$ reagent ion (Hansel et al., 1995), $NO^+$ and $O_2^+$ (Jordan et al., 2009), water clusters of $H_3O^+$ (*i.e.,* $(H_2O)_nH_3O^+$) (Breitenlechner et al., 2017), $NH_4^+$ (Zhang et al., 2018), and a few others [see (Blake et al., 2009) and references therein] have been employed. *Selected ion flow tube* (SIFT) applications have similarly utilized $H_3O^+$, $NO^+$, and $O_2^+$ ionization schemes augmented further (in certain applications) by their fast switching and combined use of simultaneous reagent ions (Smith and Španel, 2005). For similar mass spectrometric detection as in the current work, Brophy and Farmer (Brophy and Framer, 2015) have introduced a fast, switchable source for two concomitant ion mode operation and demonstrated it with the common low-pressure CIMS reagent ions acetate ($CH_3COO^-$) and iodide ($I^-$). Also, other switchable ion sources have been introduced (*e.g.,* Hearn and Smith, 2004; Agarwal et al., 2014; Pan et al., 2017), but with roughly similar utilization and performance characteristics as the ones presented above. In all these previously reported techniques the sample ionization occurs at reduced pressure, constituting sample pretreatment in drawing the atmospheric pressure material into the vacuum chambers of the MS and consequently diluting the number concentration of the sample molecules considerably (*i.e.,* by a factor of about 10 to $10^5$). This renders almost any detection method useless in targeting analytes with very low gas-phase concentrations, such as ambient aerosol precursors *sulfuric acid* (SA) and individual *highly oxidized multifunctional compounds* (HOM, 'highly oxygenated organic molecules'), which are present at around $10^5$ $cm^{-3}$ to $10^7$ $cm^{-3}$ (Eisele and Tanner, 1993; Sipilä et al., 2010; Ehn et al., 2014; Rissanen et al., 2014; Jokinen et al., 2015; Kirkby et al., 2016). Also, dilution through a small orifice and subsequent turbulent mixing of ions into the sample in the low-pressure CIMS enhances recombination loss-processes (*e.g.,* radical-radical combination reactions and wall reactions of reactive and sticky compounds) causing depletion and subsequent detection bias. The current design is virtually devoid of these issues as the ionization occurs at atmospheric pressure by an ion insertion and the sample passes the ion source unperturbed.

While the utilization of multiple reagent ions within a single CIMS apparatus offers significant benefits, to this date it has not been reported in applications in which the ionization occurs at ambient pressure. Here we introduce a significant progress in the CIMS methodology by enabling atmospheric pressure sampling and ionization with multiple consecutive reagent ions in fast repetition, and without any pre-treatment of the sampled gas mixture. When considering the detection of low-volatile, *in-situ* aerosol precursor compounds such as HOM that (i) lack analytical standards, (ii) have a range of individual detection sensitivities (Hyttinen et al., 2015; Hyttinen et al., 2018), (iii) whose transmission and (iv) fragmentation are dependent on the detailed MS settings (Heinritzi et al., 2016; Zapadinsky et al., 2019; Passananti et al., 2019) and (v) the detection is sensitive to changes in temperature (Frege et al., 2018), it becomes evident that retrieving complementary data depending on only one instrument calibration factor is extremely valuable. In the newly developed *Multi-scheme chemical IONization inlet* (MION) described here, the only changing parameter between the ionization stages is the ion specific sensitivity to the target compounds as a function of the ion-molecule reaction time - which does not differ between applications. We report the characteristics and operation principle of this new inlet and compare its performance against previously reported CIMS inlets, by coupling it into an *atmospheric pressure time-of-flight mass spectrometer* (APi-ToF-MS; Junninen et al., 2010) and measuring a multitude of previously reported oxidation products from much studied cyclohexene and α-pinene ozonolysis systems (*e.g.,* Yokouchi et al. 1985; Hatakeyama et al., 1985; Rissanen et al., 2014; Rissanen et al., 2015). To the best of our knowledge, the current type of an ambient pressure application in which the reagent ion scheme can be changed within a second time-scale by simply switching a few voltages, has not been introduced previously.

## 2 Description of the MION

A schematic of the *Multi-scheme chemical IONization inlet*, MION, (Karsa Ltd.) is presented in Figure 1. The inlet consists of an electrically grounded 24 mm *inner diameter* (i.d.) flow tube with multiple coupled ion sources (simplified two-source setup shown in Figure 1). A gas-phase stream of nitrogen or air (both of purity > 99.99999%, AGA) is enriched with the reagent ion precursor by feeding it through a saturator (or the precursor is obtained directly from a gas cylinder). The resulting reagent flows are then fed into their respective ion sources, where the reagents are ionized by a soft x-ray radiation (Hamamatsu L12535). The reagent ions are then accelerated and focused through 5 mm orifices into the laminar sample flow by electric fields. Small counterflows are applied through the orifices to prevent the mixing of the electrically neutral reagent precursors with the sample flow. Besides the ion source orifices, the flow tube design differs only minimally from tubular, making the flow pattern easy to define, the flow being essentially a flow through a circular pipe. The distance between the downstream ion source and the mass spectrometer pinhole is fixed, but the modular design of the MION allows the upstream source distance to be chosen to suit the application, with around 50 *milliseconds* (ms) minimum between ionization stages and longer times achievable by increasing the pipe length between the sources. For the experiments described in this paper the reaction time for the upstream reagent (nitrate ion, $NO_3^-$) was 300 ms, and for the downstream reagent (bromide ion, $Br^-$) 30 ms, when the total sample flow was 20 *liters per minute* (lpm).

*a)*

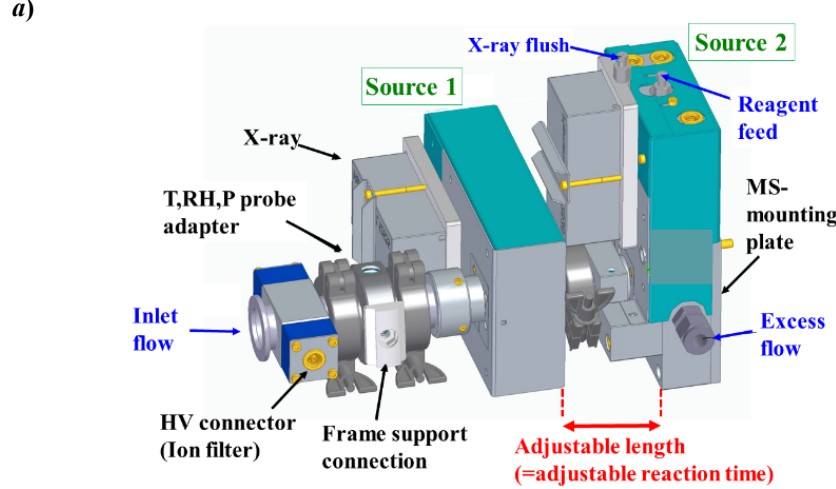

*b)*

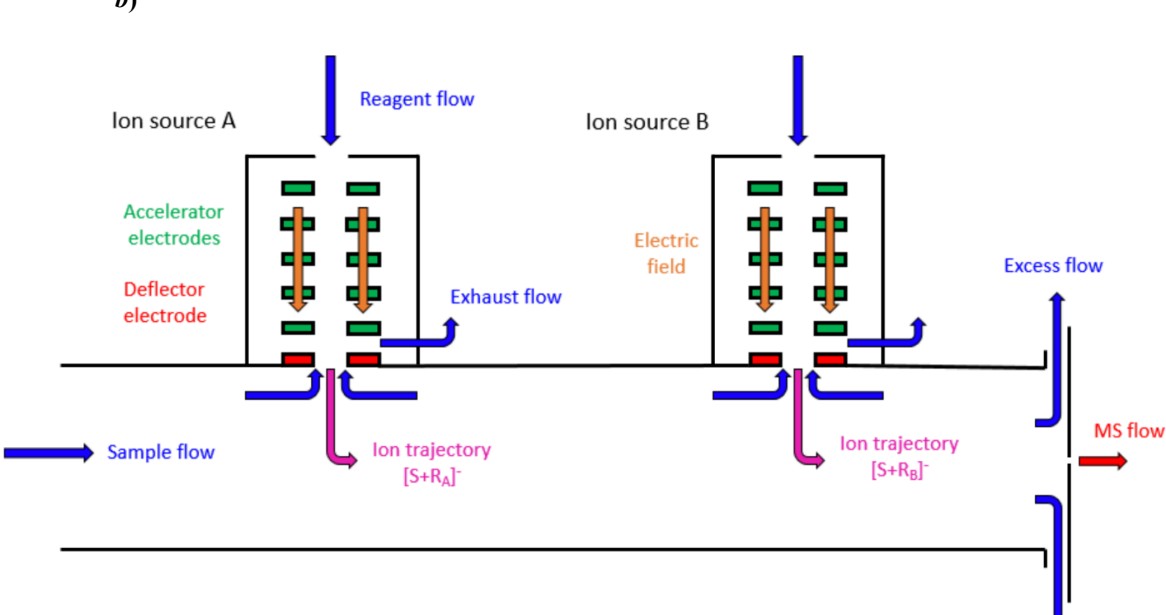

**Figure 1:** The *Multi-scheme chemical IONization inlet* (MION). *a)* A schematic description of the modular design shown with two concomitant ion sources with adjustable ion-molecule reaction time between them, coupled to an ion filter and meteorology sensor. *b)* Illustration of the gas flows and ion paths within the inlet.

### 2.1 Operating the MION

MION is operated by first turning on the counterflows for the ion sources, followed by turning on the sample flow, after
130 which the reagent feed is turned on. The sample flow is controlled by a 50 lpm *mass flow controller* (mfc), and the counterflows and reagent feeds are controlled by 100 *milliliters per minute* (mlpm) mfcs. The flow rates for the reagent feeds and counterflows depend on the compounds used as reagent ion precursors, but generally suitable counterflow (*i.e.,* the counterflow setting subtracted by the reagent feed flow setting, producing a small suction to the sample flow) for the ion sources is as little as 25 mlpm. After the flows have been chosen, the high voltage (*c.a.* 2500 V) for the ion
accelerator arrays is turned on. Lastly, to guide the reagent ions to the center of the sample flow, the ion deflector voltage (*c.a.* 250 V) for the selected ion source is turned on. If the deflector electrode is grounded, it acts as an ion filter, and the ions produced in the ion source will not enter the sample flow. Selecting an ion mode (*i.e.,* switching the ionization scheme) is as simple as turning on the deflector voltage for the desired ion source, i.e., turning on the deflector voltage in ion source A, while keeping the deflector in source B grounded, selects the ion chemistry of source A, and vice versa.
Turning off (to 0 V) both deflectors will prevent all the ions from both of the ion sources from entering the sample flow, and thus, the ambient ion mode is selected. Turning on both deflectors simultaneously will result in the downstream ion

source ionization scheme to apply, as the downstream deflector will deflect most of the ions from the upstream one. While there may be settings where multiple chemical ionization schemes can work simultaneously, the application described in this paper was not designed nor tested for that.

**2.2 Switching the ion chemistry**

The MION design enables rapid, concomitant application of multiple reagent ion chemistries, which selection and combination are mainly dictated by three variables: (*i*) target species to be ionized, (*ii*) characteristics of the reagent ion source compound, and (*iii*) the details of the ion-molecule reaction chemistry [*e.g.,* adduct formation vs charge transfer (Hyttinen et al., 2018)]. By considering these issues it is possible to choose a combination of ionization schemes that critically supplement each other, significantly increasing the chemical information obtained from the targets. For example, measuring "a complete" product distribution with an inherently unselective reagent ion (*e.g.,* bicarbonate, $HCO_3^-$) followed by a step-wise increase in chemical selectivity on further ionization stages [*e.g.,* peroxy acids by $I*H_2O^-$ (Iyer et al., 2017) and hydroperoxides by $CF_3O^-$ (Crounse et al., 2006)]. Similarly, adduct forming reagent ions (*e.g.,* $NO_3^-$) can be augmented with ions strongly participating in further ion-molecule reactions with characteristic reaction and fragmentation patterns. An ion combination which works well with the same mass spectrometer settings between the ionization schemes is highly beneficial, as in this way the ion transmission characteristics of the instrument are minimally affected, and the detected signal heights depend mainly on the individual product*reagent ion binding strengths. This detection issue can be largely avoided if the two ionization schemes work with different ionization mechanism, i.e., if the product detection sensitivity depends on the extent of fragmentation only in one mode and the other mode is barely influenced by the electric field strength change (e.g., if the first mode creates adducts and the second mode transfers charge, leading to ion-adducts and charged molecules, respectively). Moreover, the design of the MION is ideally suited for investigating the detailed influence of ion-molecule reactions and reaction times (and thus also ion-molecule reaction kinetics), enabled by using the same reagent ion precursor feed for multiple ion sources. While in principle any combination of reagent ions is possible in the MION, for this work $Br^-$ and $NO_3^-$ were selected not only due to their differing ionization characteristics and good performance under identical mass spectrometer settings, but also for their potential to offer complementary insight into the inspected VOC oxidation processes.

The fast switching between the ion modes is illustrated in Figure 2, where the signals obtained for the reagent ions (*i.e.,* $Br^-$ and $NO_3^-$) and the *total ion current* (TIC) are shown for an experiment where both ionization modes and a natural ion measurement (labeled APi in Figure 2) were utilized to measure laboratory air. When neither source is active, the system measures only natural ambient ions, which results in many orders of magnitude lower signal level. The rapid ion mode switch is completed within about a second timeframe, with minimal interference from the idle ion mode. The small nitrate ion signal measured during bromide mode indicates that some nitric acid was present in the sampled gas stream and is ionized by $Br^-$.

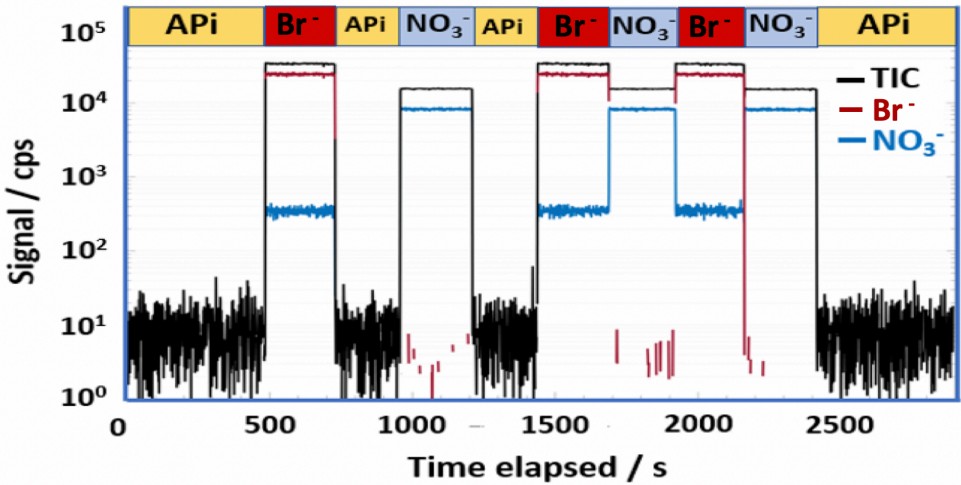

**Figure 2** An example of switching between multiple ion chemistries while sampling laboratory air. The ion modes utilized have been marked with separate colours and are further labelled above the figure with abbreviations: APi for ambient ion mode (no active ionization applied; black trace), $Br^-$ for bromide ion ionization mode (red trace), $NO_3^-$ for nitrate ion ionization mode (blue trace), and TIC is used to indicate total ion count measured (black trace). All signals shown here were retrieved by high resolution peak fitting.

## 3 Characterizing the MION – Sensitivity to gas-phase aerosol precursors

Sensitivity of the MION was inspected in two independent ways: (*i*) by calibrating its response in both ion modes to photochemically produced *sulfuric acid* (SA, $H_2SO_4$), and (*ii*) by measuring oxidation products of ozonolysis initiated autoxidation of cyclohexene and α-pinene, especially targeting HOMs (Rissanen et al., 2014; Rissanen et al., 2015). These very low-volatile, *in-situ* atmospheric aerosol precursor compounds are typically present at around $10^5$ to $10^7$ cm$^{-3}$ concentrations in the ambient gas-phase, and thus an ability to directly detect them will demonstrate the new inlet design's applicability for conducting field measurements. In addition, the SA calibration has been the standard method for estimating nitrate CIMS response to HOM detection (Ehn et al., 2014; Rissanen et al., 2015), which completely lacks any analytical standards, and, the HOM product distributions of these two prototypical endocyclic alkenes are currently the best known.

### 3.1 Sulfuric acid calibration

Figure 3 shows the determined SA calibration plots, in which the measured $H_2SO_4$ is compared to values deduced by a reaction system simulation of OH initiated $SO_2$ photo-oxidation (see the SI for details). In both ion modes the MION detects gas-phase SA with good linearity of detection and well down to atmospheric concentration levels, with common sunlit daytime values generally ranging between $10^6$ to $10^7$ cm$^{-3}$ (Eisele and Tanner, 1993; Sipilä et al., 2010). However, the sensitivities between the ion modes had about a factor of 10 difference, perhaps somewhat fortuitously, equaling the difference between the ion-molecule reaction times. The determined calibration factors were $C_{NO3-} = (1.39 \pm 0.03) \times 10^9$ cm$^{-3}$ for the nitrate mode with 300 ms ionization time, and $C_{Br-} = (1.32 \pm 0.02) \times 10^{10}$ cm$^{-3}$ for the bromide mode at 30 ms ionization, where the uncertainties refer to the statistical errors of the fits only. The overall uncertainty in the measured $H_2SO_4$ values obtained with this procedure was previously determined as 33% (Kürten et al., 2012), and the resulting uncertainties in HOM detection have been reported previously as ±50% (Ehn et al., 2014) or a factor of 2 (Berndt et al., 2015).

The obtained $C_{NO3-}$ value compares well with the lower limit values reported previously for the nitrate CIMS employing APi-ToF-MS as the detector, which range roughly from $10^9$ to $10^{10}$ cm$^{-3}$ (*e.g.,* Ehn et al., 2014; Rissanen et al., 2014; Berndt et al., 2015; Jokinen et al., 2015). For the Br$^-$ mode there are no previous values reported for SA detection. However, the $HO_2$ radical detected with the bromide ionization well at roughly few ppt concentrations in the flow reactor experiments described next, is in accordance with the observations of Albrecht et al. (Albrecht et al., 2019) and Sanchez et al. (Sanchez et al., 2016), despite the differences in the ion source operating conditions (i.e., current ambient pressure ion source in contrast to the low-pressure sources used in the mentioned previous works).

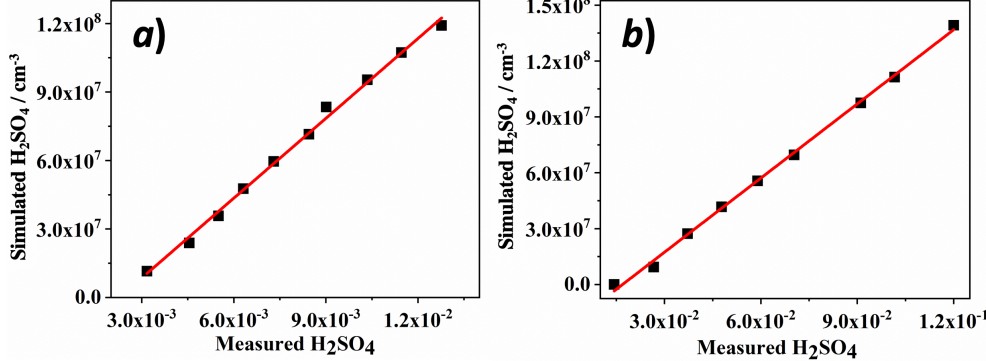

**Figure 3** Sulfuric acid ($H_2SO_4$) calibration plots showing the simulated values in molecule cm$^{-3}$ against the measured, normalized ion signals determined for *a)* Br$^-$ ion mode, and for *b)* NO$_3^-$ ion mode. For the ion signal to concentration conversion see the supplementary material.

### 3.2 VOC oxidation products

Cyclohexene ($C_6H_{10}$) and α-pinene ($C_{10}H_{16}$) ozonolysis initiated autoxidation were investigated to inspect the applicability of the new inlet design to detect various reaction products with differing oxygen content, and to compare its performance to previously reported results obtained with other CIMS inlets (*e.g.,* Ehn et al., 2014; Rissanen et al., 2014; Berndt et al., 2015; Mentel et al., 2015). The experiments were performed in quartz flow tube reactors (2.44 cm i.d. and 80 cm length or 7.7 cm i.d. and 120 cm length) under atmospheric conditions with an inlet flow of 20 lpm for the MION, resulting in 2 to 10 second reaction times. The hydrocarbon precursor, at around 100 ppb in nitrogen ($N_2$), was mixed with the bath gas air ($N_2 + O_2$) and ozone ($O_3$, *c.a.* 50 ppb), few centimeters upstream of the flow reactor. The introduction of ozone into the flow reactor containing the hydrocarbon precursor resulted in apparent instantaneous formation and subsequent

detection of various HOM products, implying that rapid autoxidation of the endocyclic precursors took place, and that the new inlet design can detect these *in-situ* aerosol precursors present at very low concentrations [*i.e.,* fractions of ppt to several ppt for individual HOM as reported previously for short reaction time conditions (Jokinen et al., 2015; Berndt et al., 2016)]. The monomer and dimer HOM products detected here are covalently bound distinct molecules and do not result from ion-ion or ion-molecule reactions in the atmospheric pressure ionization inlet. In the recent literature, a considerable effort has been invested to unambiguously explain their origin and identity (see a recent review by Bianchi et al. 2019 and references therein). More details of the experimental setup and conditions can be found in the SI chapter S2.

Typical spectra obtained with the MION during an α-pinene ozonolysis experiment in both ion modes are shown in Figure 4 (for cyclohexene spectra see the SI); the bromide spectra have been shifted 17 mass units to illustrate common compositions on top of each other in the spectra, and the mass peak ranges have been divided into the reagent ion, monomer and dimer mass ranges (see Figure 4 and Figure S3 in the SI). Inspecting the obtained spectra, it becomes immediately evident that only the bromide ionization retrieves few of the least-oxidized reaction products (but to a large extent also the further oxidized compounds), whereas the extremely selective $NO_3^-$ ionization is more sensitive to the highest oxidized species. However, in the present work, a 10 times longer ion-molecule reaction time was used for the $NO_3^-$ mode, and thus the sensitivity to the higher oxidized products is likely further augmented by this longer ionization time, which generally leads to increased sensitivity for the strongest bound reagent*product adducts, usually to the highest oxidized reaction products (Hyttinen et al., 2015; Hyttinen et al., 2018). Naturally, the sensitivity changes as a function of the ion-molecule reaction time and each ion-source distance needs to be calibrated to obtain their detailed response. In the MION setup, the ion-molecule reaction time is easily adjustable, and thus is ideally suited to study these influences and even ion-molecule reaction kinetics. Nevertheless, the charging mechanism between these reagent ions differ; in nitrate ionization the analyte and $HNO_3$ compete for $NO_3^-$ whereas, in bromide ionization, $Br^-$ directly forms adducts with the analyte, *i.e.,* the former is a ligand switching reaction while the latter is a direct adduct formation (Hyttinen et al. 2015; Hyttinen et al. 2018), leading to inherently less selective ionization by $Br^-$. Additionally, bromide ionization results in significantly more organic ion products devoid of the reagent ion, implying that other ion-molecule reactions are also involved in $Br^-$ ionization [*e.g.,* deprotonation (Hansel et al., 2015; Breitenlechner et al., 2017) or dehydroxylation (Mielke et al., 2012; Iyer et al., 2017)] than simply adduct formation (see the SI for further details). In contrast, $NO_3^-$ was only observed to deprotonate a few dicarboxylic acids and SA reported in previous publications (Eisele et al., 1993; Rissanen et al., 2014). The organic ions were always minor in comparison to the most prominent product peaks.

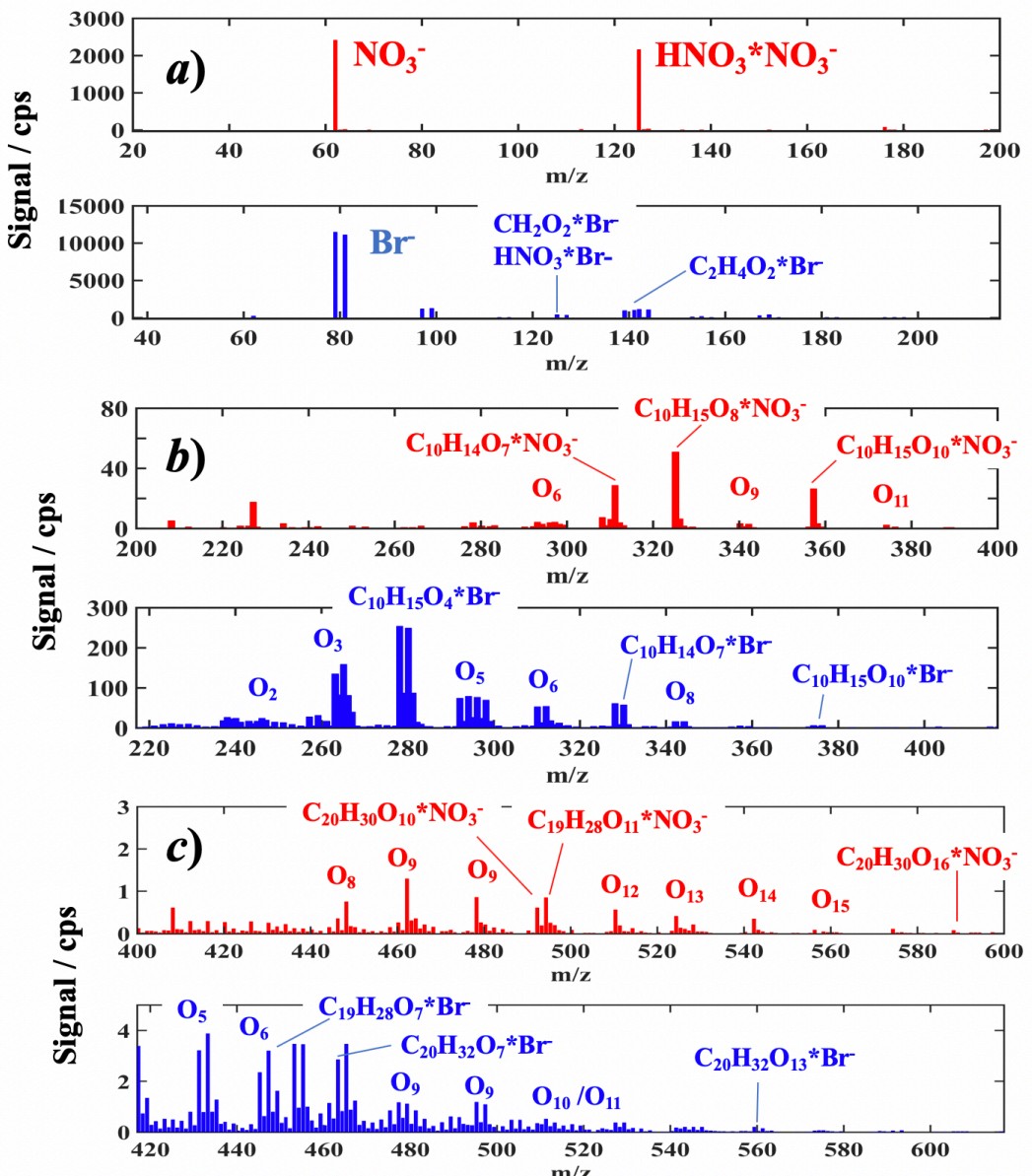

**Figure 4** Example MION spectra of organic oxidation products obtained from α-pinene ozonolysis experiments shown with a common product mass axis for both ion modes, *i.e.,* the Br⁻ spectrum (blue) is displaced by 17 Th (=difference between reagent ion Br⁻ and $NO_3^-$ masses) to overlap the same composition products horizontally. **a)** Illustrates the reagent ion spectra, **b)** the monomer range (*i.e.,* oxidation products which have the same number or less carbon atoms), and **c)** the dimer range (*i.e.,* oxidation products with about two times the carbon number of α-pinene), respectively. For a few most prominent products the explicit compositions are given, whereas for most of the detected peak groups only the average number of oxygen atoms is indicated; in dimer range the products with adjacent number of carbon and oxygen atoms start to overlap.

A critical aspect of using chemical ionization techniques is the sufficient availability of reagent ions. That is, when the analyte concentrations are too high, the observed reagent ion signal is considerably depleted. Under these conditions, the detection is rendered qualitative and it is not possible to determine the analyte concentrations. In the opposite situation, with too high primary ion production, there is a potential for secondary ion-ion reactions contributing to the measured apparent product ion signals. These finite limits exist for every chemical ionization inlet and are heavily dependent on the primary ion production rate and geometry of the ion source. For the current inlet system, we did not observe increased production of "dimer" or fragment products nor did we observe significant depletion of reagent ions, although a minor reduction in bromide reagent signal is evident when the highest concentration mixtures were sampled (at about 50 ppb of $O_3$ and 500 ppb of α-pinene; see Figure 5 and Figure S1). However, this small reduction in bromide reagent ions did not deteriorate the linearity of the detection scheme (see Figure S2). Furthermore, the measured product distributions closely resembled those that we have determined previously with an 'Eisele-Tanner-type' inlet (Eisele and Tanner, 1993), verifying that if such influences were present they were minor at best.

The concomitant application of $NO_3^-$ and $Br^-$ ionization is enabled by their comparable adduct binding strengths with oxidized organic molecules (Hyttinen et al., 2018). This means that both product-adducts are seen at similar efficiency, with the same mass spectrometer settings, and that strong adduct fragmentation would be seen in both ion modes if present. With very different product*adduct binding strengths, the least strongly bound adducts are likely lost to fragmentation in the APi section of the MS (Zapadinsky et al. 2019; Passananti et al. 2019). However, if the charging mechanism between the two reagent modes differ (*e.g.,* one forms adducts and the other transfers charge), then they are likely applicable together, and are largely unaffected by the mass spectrometer settings.

The fast switching of peroxy radical ($RO_2$ and $HO_2$) reagent*product adduct detection is illustrated in Figure 5, where the ion signals obtained for the reagent ions and several α-pinene oxidation products are shown for a flow tube experiment conducted in air bath gas, at a 10 seconds residence time. High reactant concentrations were used and the product signal levels are observed to change according to the VOC load of the flow reactor, marked in the figure by vertical dashed lines. The rapid ion mode change is completed within about one second timeframe and is seen, for example, in the time traces measured for the prominent, highly-oxidized peroxy radical formed by autoxidation, $C_{10}H_{17}O_{17}$. This compound was detected well in both ion modes. Here, switching of the ion mode only shifts the mass of the product peak in the spectrum, with associated shift in signal intensity due to differences in the absolute detection sensitivities and total ion counts between the ion modes. In contrast, the $HO_2$ radical and the primary ozonolysis derived peroxy radical, $C_{10}H_{15}O_4$, are detected only in the $Br^-$ mode and thus disappear from the spectrum when nitrate mode is utilized. Similarly, few of the most oxidized reaction products were prominent only in the $NO_3^-$ spectra (Figure 4). As seen in Figure 5, some $HNO_3$ was present in the sampled gas flow during the whole experiment, as the $NO_3^-$ signal shows up during $Br^-$ stages. At the highest concentration experiments (at around 16:45), also small product signals were retrieved for the highly-oxidized peroxy radicals attached to a nitrate reagent ion. Although this amount was enough to generate measurable product ion signals, it did not deteriorate the bromide ion detection (note that $Br^-$ will transfer its charge to $NO_3^-$ very efficiently and leads to strong bromide ion signal depletion, which was not observed).

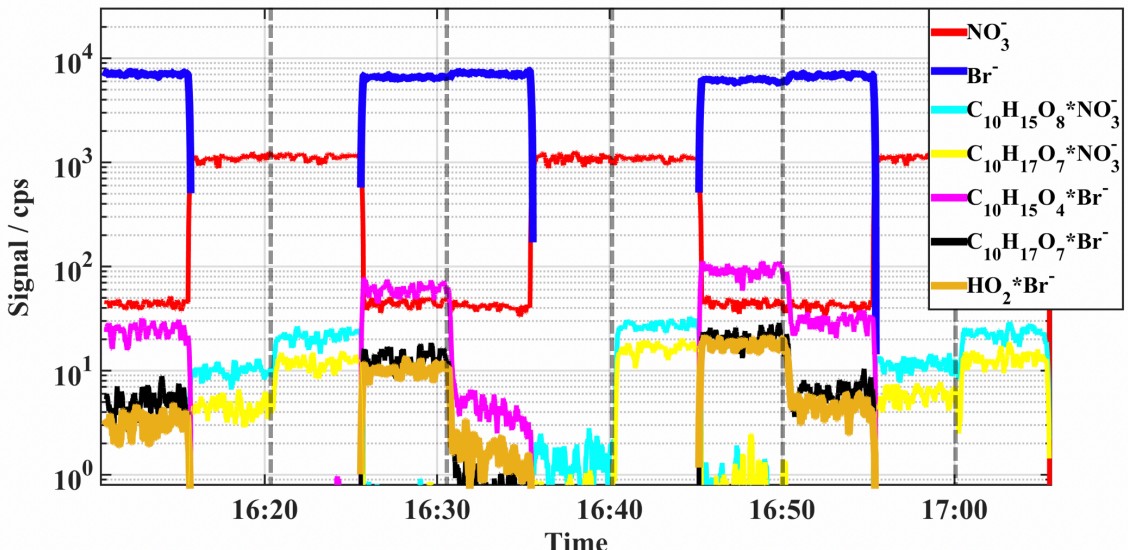

**Figure 5** Time traces measured for the reagent ions and selected α-pinene oxidation derived peroxy radicals illustrating the fast switching between reagent ion modes; the highly oxidized $RO_2$ ($C_{10}H_{17}O_7$) is retrieved well by both ion modes, whereas, the primary peroxy radicals $HO_2$ and $C_{10}H_{15}O_4$ were detected only with $Br^-$ ionization. The changes in the product signal levels are due to different hydrocarbon loads in the flow reactor, and the changes have been marked with vertical dashed lines. At the highest concentration experiments (at around 16:45; α-pinene about 500 ppb), small but measurable signals were retrieved also for the nitrate ion clustered products.

**4.1 Comparison to previously reported cyclohexene and α-pinene HOM product distributions**

The cyclohexene ozonolysis HOM product distribution reported previously in Rissanen et al. (Rissanen et al., 2014), Berndt et al. (Berndt et al., 2015) and Mentel et al. (Mentel et al., 2015) is well recorded by the $NO_3^-$ ion mode of the MION, and even extends to the somewhat less oxidized reaction products. This is likely due to the different ion injection process in the current inlet design which could potentially concomitantly decrease the sensitivity for the highest oxidized compounds (Hyttinen et al., 2015). However, the observed sensitivity to the highest oxidized species is similar within the measurement uncertainties to our previous works (*e.g.,* Rissanen et al., 2014; Rissanen et al., 2015) and all the previously reported HOM by $NO_3^-$ ionization are detected with the MION setup too.

In the bromide mode, also the least oxidized reaction products down to 2 O-atoms were recorded, which amply illustrates the benefit from the multiple ion operation. To the best of our knowledge, this is the first-time bromide ionization has been used to detect gas-phase cyclohexene and α-pinene ozonolysis products and thus a direct comparison to previous results is not possible. Nevertheless, the $Br^-$ ionization retrieves even earlier reaction products in cyclohexene oxidation than those reported with the acetate ($CH_3COO^-$) (Berndt et al., 2015; Hyttinen et al., 2017) and iodide ($I^-$) (Mielke et al., 2012; Iyer et al., 2017) reagent ions. For α-pinene, in addition to the HOMs reported previously by $NO_3^-$ ionization (*e.g.,* Ehn et al., 2014; Rissanen et al., 2015), bromide picks up several products with compositions matching the well-known, abundant, early generation oxidation products (*e.g.,* pinic acid $C_9H_{14}O_4$) (Ma et al., 2007) but which could also result from isomeric product compounds detected at the same exact mass. Additionally, ozonolysis of endocyclic alkenes produces a range of smaller carboxylic and peroxy acids as primary and secondary reaction products (Johnson and Marston, 2008), some of which were also detected in the current experiments with high sensitivity (*e.g.,* based on the current alkene + $O_3$ reaction rates, these products should be present at around maximum 1 ppb concentration with the long reaction time and high VOC loading experiments). Formic acid (HCOOH) and acetic acid ($CH_3COOH$) were detected in the experiments, together with peaks having matching compositions to higher carboxylic acids, but which do not have unambiguous molecular compositions for definite mass specific identification. In addition, the strong acids nitric acid ($HNO_3$) and sulfuric acid ($H_2SO_4$) were also detected in the bromide mode, the latter used in calibration, and the former being a by-product of unrelated $NO_x$ experiments performed with the same inlet system.

### 4.2 Comparison to previously reported CIMS applications using multiple reagent ions

The MION inlet represents a significant improvement in measuring analytes at atmospheric pressure by employing multiple complementary ion chemistries. As mentioned above, to our knowledge, this is the only application in which the ionization occurs under ambient conditions and is completely devoid of any sample pre-treatment. Perhaps closest to the current application is the two-ion-mode setup introduced by Brophy and Farmer (Brophy and Farmer, 2015), in which the ionization scheme is likewise changed by a simple ion optics voltage switching, but which operates under reduced pressures (roughly at 0.1 Atm), inherently lowering the sensitivity to the electrically neutral target compounds. In low-pressure inlets, the number concentration (i.e., the number of molecules/unit volume) of electrically neutral sample material is invariably diluted when pulled into the vacuum of the ionization chamber. While the low-pressure systems suffer from this dilution they generally generate significantly higher ion counts and as such do not experience as serious reagent ion depletion as the ambient pressure sources. However, when targeting neutral analytes with very low ambient concentrations (e.g., HOMs and other in-situ aerosol precursors) it is beneficial to avoid diluting the already low gas-phase concentration, and instead, apply atmospheric pressure ionization before the vacuum chambers. Crucially, the MION is not limited to two-ion-mode operation described in the present paper and, due to its modular design, can accommodate as many reagent ions as feasible, only limited by space, utilizable ion-molecule reaction times and other physicochemical constraints set by the reagent precursors.

The recently developed PTRMS instruments (Breitenlechner et al., 2017; Krechmer et al., 2018) can, in principle, rival the multi-ion operation due to their wide detection range for different oxidized states, and their applicability for direct ambient pressure sampling. However, in practice, the product analysis is tedious due to the inherently low selectivity of the common $H^+$ transfer reagents, resulting in vast number of overlapping compounds. Yet again, the ionization in PTRMS instruments occurs at low pressures (commonly around 1 mbar), resulting in loss of sensitivity discussed above.

### 5. Conclusions

Atmospheric pressure chemical ionization mass spectrometry utilizing multiple reagent ion chemistries is a powerful analysis method for probing various gas-phase processes. The new type of inlet design introduced here enables selective inspection of gas-phase product distributions, allowed by rapid switching of chemical selectivity *via* specific reagent ion chemistries. This principle was demonstrated by detecting various previously reported oxidation products of cyclohexene and α-pinene applying $Br^-$ and $NO_3^-$ ionization schemes in parallel. The sensitivity of the MION was observed to rival the previously reported CIMS inlets in detecting HOMs and sulfuric acid, exemplifying its applicability for field deployments. While the current setup was optimized for two-ion-mode operation, the modular construction of the MION enables it to, in principle, operate with multiple, unlimited ionization stages in fast repetition and without any instrument or sample pre-treatment, the ion mode being changed by a simple switching of a few low voltage settings.

**Acknowledgements**
MPR is grateful for the support from the Academy of Finland (Grant numbers 299574 and 326948).

## Supporting Information

The Supporting Information material includes sections S1-S6: S1 Abstract – description of the supporting material, S2 Experimental setup and conditions, S3 Ion signal to concentration conversion, S4 Calibration measurement, S5 Product distributions – specific reaction products, S6 Organic ion peaks devoid of the reagent ion. Supporting Figures S1-S5; example MION spectra.

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
