# Peer review of "Multi-scheme chemical ionization inlet (MION) for fast switching of reagent ion chemistry in atmospheric pressure chemical ionization mass spectrometry (CIMS) applications"

_Atmospheric Measurement Techniques, 2019_

## Referee Comment (RC1) · Anonymous Referee #1 · 14 May 2019

This manuscript presents a system for switching reagent ions in a chemical ionization mass spectrometer, with application in the field of atmospheric chemistry. The authors discuss the use of the system with bromide and nitrate ions to investigate gas-phase oxidation chemistry. The authors present interesting data on bromide ionization chemistry, particularly the use for measuring sulfuric acid. The authors contrast their new ion source to the literature. Overall, this paper will be a useful contribution to the literature and is appropriate for this journal. However, there are a few specific experiments or pieces of information missing that are essential for proving that the multi-reagent ion

switching system works. I recommend several major revisions.

Major comments. 1.My biggest concern with this ion setup with atmospheric pressure is the potential for secondary chemistry and (unexpected) cluster formation. Along with that is a concern about titration of reagent ions when looking at complex, high concentration mixtures. These are challenges that must be addressed for publication. The key question is: in the mass spectra shown in Figure 4, how do you know that these peaks are the result of only ion + analyte adduct formation, and not multi-component clusters (i.e. formation of dimers in the reaction chamber as a result of sequential collisions)? If you apply different field strengths to the resulting ions, you will see the adducts fall apart, and be able to gain some insight into the mechanisms (see work by Lopez-Hilfiker on iodide CIMS or Brophy on acetate CIMS). The authors refer to formation of monomers vs dimers - but do not investigate whether these products are real, or the result of ion-molecule reactions in the instrument itself. The potential false production of dimers in the atmospheric pressure source seems challenging. Further, the number of ion-molecule collisions that will occur in the MION (i.e. calculate the mean free path and compare to the size of the ion source) suggests that secondary ionization and titration of reagent ions will be a challenge. The authors can demonstrate that titration of reagent ions isn't a problem by showing the time series of the reagent ion and total ion count during an experiment in which zero air flows into the instrument and then is rapidly switched to a complex mixture from a flow reactor. If the reagent ion signal decreases, then there is evidence of titration. This doesn't necessarily mean that the MION isn't useful, but it is important to show the limitations of the system.

2. The two reagent ion sources clearly have different reaction times based on the diagram in Figure 1, so what is the implication of these difference? The authors need to discuss how different sensitivities or mass spectra are if they run the same reagent through the two reagent ion sources and switch between them on a constant flow of a complex mixture (e.g. output from a flow reactor). How similar (or different) are the resulting spectra and sensitivities? Figure 2 shows that there is some sort of carryover:

the nitrate signal is larger during bromide ionization than it is when neither ion source is active.

Other Comments

1. Line 242. This paragraph makes no sense: the authors claim that nitrate and bromide ionization have similar adduct binding strengths, and this somehow means that the two ionization techniques can be used. Why is this the case? Why can't quite different reagent ions be used in the system? Later in the paragraph, the authors seem to say that one can use reagent ions with different ionization mechanisms, so the authors need to clarify their point.

2. The authors use calibrations of nitrate and bromide CIMS for sulfuric acid to prove that their instrument is capable of measuring this species. However, it is not clear if these experiments were done in the 'rapid switching' mode that is the core of the paper. In fact, the inset in Figure 3 suggests that the calibration was done independently for the two reagent ions. If this is the case, then this section does not support the central claim and focus of the manuscript that the switching reagent ion system provides quantitative measurements of sticky compounds like H2SO4! The measurements must be done in switching reagent mode. Please show the timeseries along with the calibration curves (i.e. I like the setup of Figure 3 - it just needs to demonstrate that these curves hold under the actual reagent ion switching setup, and at different relative humidities).

3. Sulfuric acid detection by bromide reagent ions is a constructive direction - but needs more analysis to support the claims. Specifically, I would like to see: is there a dependence on relative humidity? And two what extent does the system form clusters vs bare ions?

4. Figure 4: I think the authors intend to show that peaks in the spectra are oxygenated organics as demonstrated by labels of O5 / O6 / O7 / etc. Are these peaks actually CxHyOz=5,6,7 or are they truly O5- ions? Please label according to actual chemical formulae.

5. I am confused by the details in Section 2.1. The authors refer to ion accelerator arrays and an ion deflector voltage. Where are these? It would be helpful to add these to the Figure 1 schematic.

6. From the schematic and description in Schematic 1, it sounds like the reagent flow is constant through both ion sources – what switches are the electric fields that ionize the reagent ions. This is an elegant solution to rapidly switching reagent ion flows. However, it does raise the question of the extent to which the (unionized) reagent flow impacts the ion-molecule reactions and chemistry in the CIMS inlet. Is there an interference from the (unionized) reagent in the mass spectra? For example, if you using a bromine precursor as a reagent flow in the first ion source, but are under nitrate ionization, do the nitrate reagent ions react with the brominated compounds? I imagine there are some combinations of reagent ion precursors where this could be an issue. The authors should (quantitatively) comment on the potential of one (unionized) reagent precursor to titrate the (ionized) reagent ion or compete with the intended reagent ion + analyte molecule reactions. 7. Line 305. The authors suggest that operating CIMS at low pressures inherently reduces signal by dilution. This makes no sense to me – in fact, operating ion-molecule reaction chambers at reduced pressure can present the advantage of avoiding titration of the reagent ion and minimizing secondary ionization reactions. The authors will need to explain their point in detail.

Technical comments

1. line 82, the authors claim that the work 'represents a significant leap in the CIMS methodology'. I disagree with the term significant 'a significant leap' and suggest it be replaced with 'progress'. This work represents progress – but 'significant leap' is an over-statement.

2. The authors note on line 96 that this is the first time in which the reagent ion scheme can be switched 'quantitatively'. This makes no sense: the reagent ion systems of the previous citations are all quantitive, but I do not understand what is 'quantitative' about

the reagent ion switching itself. Please remove the sentence or explain what you mean by this. Line 159 again refers to a 'quantitative ion mode switch', which just doesn't make sense (the switch can't be quantitative)?

3. Use of italics to emphasize points is inappropriate in research articles and should be reserved for variables or Latin terms (e.g. 'vice versa' or 'in situ')

4. Figure 5 should be labeled with what is going on (the text says that the O3 is switching – please indicate the levels so the measurements could be taken in context

5. Line 101 should read "A schematic"

6. Figure 2 caption is inadequate to understand the figure. I think that the colored traces represent mass spectral signals for the Br- ion or NO3- ion? (high resolution or nominal mass?) And then the terms at the top of the figure are the reagent ion mode that is activated? But what is the inflow air comprised of? Room air? Standards? Chamber air?

---

## Referee Comment (RC2) · Anonymous Referee #2 · 3 Jun 2019

General Comments

The work presents a useful chemical ionization inlet setup that can be used by a growing number of research groups that use CIMS and similar techniques. In general, the manuscript is well written and the results are clearly presented. The results of sulphuric acid calibration and the comparison of the spectra from $\alpha$-pinene and cyclohexene ozonolysis for both ion modes are promising. However, as a manuscript in AMT, I would expect more technical details offered, as well as more related discussion. I suggest the manuscript can be considered for publication after clarifying following

issues:

1. I am particularly concerned with the performance of Br- CIMS at atmospheric pressure. In both works mentioned (Albrecht et al. and Sanchez et al.), Br- CIMS was conducted at low pressure as opposed to the atmospheric pressure used in this study. The authors compared the HO2 radical concentration detected in this study with the observations from these studies and got the conclusion that the sensitivity is similar (Page 4, line 194-195). Meanwhile, the authors also emphasized the advantages of an atmospheric pressure application. Please explain more about the reaction time, sensitivity, ion-molecule collision frequency, ect. and what the advantages are.

2. The comparison of spectra (Figure 4) was actually impressive. However, I don't think it is very appropriate to cluster peaks using oxygen numbers as did in figure 4. I would expect e.g. C9HyO5 and C10HyO4 are observed by Br- CIMS, but both belong to the same cluster "O4" marked in Figure 4, which is definitely not correct. It is also not clear (for both Figure 4 and Figure S2), are there any (exactly) common compounds? I suggest to label some major compounds or give a list.

Specific comments

1. Page 3, line 132-138: The description of switching between two modes is relative short and too simple. Please give more details. Another question: are there any different settings of APi region for these two modes?

2. Page 7, Figure 5: Since most of the compounds were not exclusively detected in only one mode, add some explanation of e.g. why C10H15O10*Br- was detected in NO3- mode when O3 concentration was changed (?, 16:40-16:45).

3. Overall, I feel some of the necessary experiment description is missing, and it causes difficult to follow the paper. E.g., the description of the experiment for Figure 5 is not clear enough.

Technical Comments

1. Page 2, line 103: "air", what type of air?

2. Page 4, Figure 2: Use the same colour for Br- signal and the legend.

3. Page 5, line 195: Albrecht et al. (2018) -> (2019).

4. Page 5, Figure 3: Unit is missing.

---

## Author Comment (AC2) · 5 Sep 2019

The comment was uploaded in the form of a supplement:
https://www.atmos-meas-tech-discuss.net/amt-2019-159/amt-2019-159-AC2-supplement.pdf

---

## Author Comment (AC1)

Response for the review comments on the AMT manuscript: "Multi-scheme chemical ionization inlet (MION) for fast switching of reagent ion chemistry in atmospheric pressure chemical ionization mass spectrometry (CIMS) applications" by Rissanen, M. P. et al.

First and foremost, we want to thank the reviewers for their helpful comments in improving our manuscript. Below, you will find a detailed account of the reviewer concerns (in italics), our replies to these concerns, and any changes and additions made to the text.

**Anonymous Referee #1**

**Received and published: 14 May 2019**

This manuscript presents a system for switching reagent ions in a chemical ionization mass spectrometer, with application in the field of atmospheric chemistry. The authors discuss the use of the system with bromide and nitrate ions to investigate gas-phase oxidation chemistry. The authors present interesting data on bromide ionization chemistry, particularly the use for measuring sulfuric acid. The authors contrast their new ion source to the literature. Overall, this paper will be a useful contribution to the literature and is appropriate for this journal. However, there are a few specific experiments or pieces of information missing that are essential for proving that the multi-reagent ion switching system works. I recommend several major revisions.

**Major comments.**

My biggest concern with this ion setup with atmospheric pressure is the potential for secondary chemistry and (unexpected) cluster formation. Along with that is a concern about titration of reagent ions when looking at complex, high concentration mixtures. These are challenges that must be addressed for publication.

**Author reply:**

Our results do not indicate secondary ion chemistry affecting measured ion distributions to any significant extent. This is especially evident from the close match of the observed product distributions obtained with the well characterized and much used 'Eisele-Tanner-type' inlet originally built in University of Colorado, Boulder, and later slightly modified and sold by Aerodyne Research. In case of significant secondary ion chemistry, which is caused by "too many ions in a too tight space", an increase in related products (e.g., fragments, oligomers) would be expected due to unwanted ion-ion chemistry on top of the ion-molecule reactions used for charging. No indication of secondary ion chemistry was obtained and this has now also been explicitly stated in the text (see below). [Furthermore, additional evidence is provided from another manuscript preparation for review purposes only.]

The somewhat opposite behavior, the titration of reagent ions (or rather depletion of reagent ions as titration is commonly reserved for another meaning) is a common problem with atmospheric pressure chemical ionization (APCI) methodology. Thus, it is always important to choose the right reagent ion source to suite your application. The more unselective your reagent ion is the more likely you are to deplete your reagent ions due to too large concentration of ionizable gas-phase products. Every chemical ionization setup has this finite limit as it is not possible to generate infinite number of ions in any conceivable ionization source. Figure 5 shows a common flow tube measurement conducted with high reactant concentrations, which had only a minor influence on the bromide reagent ions during the highest concentration experiments (i.e., with the highest reactant loading and correspondingly highest product signals; see Figure S1 below) and negligible influence on the nitrate reagent signal. However, the minor dip observed in bromide signal

did not deteriorate the linearity of the detection as is shown in Figure S2 below. Furthermore, as a response to the reviewer concern, this issue is now handled with more detail in the main text and in the SI. Also, these new figures were added to the SI to elaborate on the issue.

New SI Figures S1 and S2 and their captions:

**Figure S1** Linear scale duplication of the manuscript Figure 5; the small dip observed in bromide reagent ion signals when sampling high concentration mixtures. As can be seen from Figure S2, this small decrease did not deteriorate the linearity of the detection.

**Figure S2** Determined products signal levels of the peroxy radicals in manuscript Figure 5 shown as a function of the  $\alpha$ -pinene and ozone reaction rate, demonstrating the linearity of the detection in measuring high concentration gas mixture.

Addition to manuscript text (Page 7, Line 266): A critical aspect of using chemical ionization techniques is the sufficient availability of reagent ions. That is, when the analyte concentrations are too high, the observed reagent ion signal is considerably depleted. Under these conditions, the detection is rendered qualitative and it is not possible to determine the analyte concentrations. In the opposite situation, with too high primary ion production, there is a potential for secondary ion-ion reactions contributing to the measured apparent product ion signals. These finite limits exist for every chemical ionization inlet and are heavily dependent on the primary ion production rate and geometry of the ion source. For the current inlet system, we did not observe increased production of "dimer" or fragment products nor did we observe significant depletion of reagent ions, although a

minor reduction in bromide reagent signal is evident when the highest concentration mixtures were sampled (see Figure 5 and Figure S1). However, this small reduction in bromide reagent ions did not deteriorate the linearity of the detection scheme (see Figure S2). Furthermore, the measured product distributions closely resembled those that we have determined previously with an 'Eisele-Tanner-type' inlet, verifying that if such influences were present they were minor at best.

The key question is: in the mass spectra shown in Figure 4, how do you know that these peaks are the result of only ion + analyte adduct formation, and not multi-component clusters (i.e. formation of dimers in the reaction chamber as a result of sequential collisions)? If you apply different field strengths to the resulting ions, you will see the adducts fall apart, and be able to gain some insight into the mechanisms (see work by Lopez-Hilfiker on iodide CIMS or Brophy on acetate CIMS). The authors refer to formation of monomers vs dimers - but do not investigate whether these products are real, or the result of ion-molecule reactions in the instrument itself. The potential false production of dimers in the atmospheric pressure source seems challenging.

**Author reply:** The issue of atmospheric pressure ion-molecule reactions contributing to the measured HOM products has been addressed frequently in the recent literature with the conclusion it does not play any significant role (see for example Bianchi et al. 2019 and references therein). This has been crucial in showing that HOM are real covalently bound molecules that contribute to in-situ secondary aerosol formation, and, are not charged clusters formed by the atmospheric pressure ionization method. Arguably, one of the most crucial bits of information precede the autoxidation studies and came from the observation of similar monomer and dimer HOM products in naturally charged ions in a boreal forest by Ehn, et al. 2010 (note the very low concentration of ambient ions) and only afterward nitrate ionization was used for neutral HOM measurements based on the analogy of natural ion measurements (Jokinen, et al. 2012; Ehn, et al. 2012). Furthermore, it should be noted that these dimers have been observed previously also with the low-pressure sources discussed by the reviewer (see e.g., Mohr et al. 2017). We emphasize that no significant differences, for example, amplified "dimer" production, were observed between product distributions determined in our previous and in this work using different inlet design, as was already explained above.

It must also be noted that in experiments where products disappear from the mass spectra due to increasing field strength does not necessarily indicate if you're looking at a three-body cluster or potentially just a charged molecule, unless you unambiguously record and quantify all the fragment signals. However, this is not possible, as, regardless of whether you are observing a many body clusters or a charged molecule, only one entity can carry away the charge, leaving the other fragments neutral and undetectable. Thus, it is not possible to unambiguously say if you are detecting break-up of charged molecules or molecular clusters. We would also like to emphasize that the current manuscript is dedicated to introducing the new inlet design capable of multiple ion operation and not for comparison on specific ionization techniques. We are currently working on characterizing bromide vs nitrate ionization experimentally and theoretically in another project, with a different set of authors, and which intricacies certainly deserve for a dedicated publication of its own.

Addition to manuscript text (page 5, line 231): The monomer and dimer HOM products detected here are covalently bound distinct molecules and do not result from ion-ion or ion-molecule reactions in the atmospheric pressure ionization inlet. In the recent literature, a considerable effort has been invested to unambiguously explain their origin and identity (see a recent review by Bianchi et al. 2019 and references therein).

Further, the number of ion-molecule collisions that will occur in the MION (i.e. calculate the mean free path and compare to the size of the ion source) suggests that secondary ionization and titration of reagent ions will be a challenge. The authors can demonstrate that titration of reagent ions isn't a problem by showing the time series of the reagent ion and total ion count during an experiment in which zero air flows into the instrument and then is rapidly switched to a complex mixture from a flow reactor. If the reagent ion signal decreases, then there is evidence of titration. This doesn't necessarily mean that the MION isn't useful, but it is important to show the limitations of the system.

**Author reply:** These concerns were already replied above; the depletion of the reagent ion is a common problem in atmospheric pressure ionization techniques, and is highly dependent on the ion chemistry used. While the reagent ion signal is not equivalent of comparing with the total ion count, easily the biggest portion of the ion signal is generated by the reagent ions, and as shown in Figures 5, S1 and S2, the change in reagent ions is minimal while sampling a complex gas mixture. We emphasize again that our manuscript is first and foremost an introduction to a new method capable of fast switching between different ionization chemistries. Scrutinizing the limitations of the ionization chemistries selected for demonstration purposes is out of the scope of the manuscript. [*As stated above, additional evidence is provided from another manuscript preparation for review purposes only.*]

The two reagent ion sources clearly have different reaction times based on the diagram in Figure 1, so what is the implication of these difference? The authors need to discuss how different sensitivities or mass spectra are if they run the same reagent through the two reagent ion sources and switch between them on a constant flow of a complex mixture (e.g. output from a flow reactor). How similar (or different) are the resulting spectra and sensitivities?

**Author reply:** The influence of different ion source reaction times, coupled to the ongoing oxidation reaction in the flow reactor, is not completely straightforward. The two factors to consider are the (i) longer oxidation reaction time for the downstream source, and (ii) the longer ion-molecule reaction time for the upstream source. The longer oxidation time favors the highest oxidized reaction products as they simply had more time to form, but so does the longer ion-molecule reaction time as the highest oxidized reaction products are generally stronger bound, and longer ion-molecule reaction time means approaching the thermochemical equilibrium - and thus favors the strongest product clusters (see for example: Hyttinen et al. 2015; Hyttinen et al. 2018). The point (i) will be of minor importance if the charging times are relatively low in comparison to the reaction times. Naturally, the sensitivity changes as a function of the ion-molecule reaction time, and thus each ion-source distance needs its own calibration to obtain their response to the targeted products. In the MION setup, the ion-molecule reaction time is adjustable, as explained in the manuscript text, and thus is ideally suited to study these influences and even ion-molecule reaction kinetics. In response to the reviewer concern, a mention of this was added to the manuscript text.

**Changes to manuscript text (page 6, line 242):** However, in the present work, a 10 times longer ionmolecule reaction time was used for the NO3- mode, and thus the sensitivity to the higher oxidized products is likely further augmented by this longer ionization time, which generally leads to increased sensitivity for the strongest bound reagent\*product adducts, usually to the highest oxidized reaction products (Hyttinen et al., 2015; Hyttinen et al., 2018). Thus, the sensitivity changes as a function of the ion-molecule reaction time and that each ion-source distance needs to be calibrated to obtain their detailed response. In the MION setup, the ion-molecule reaction time is easily adjustable, and thus is ideally suited to study these influences and even ion-molecule reaction kinetics.

Figure 2 shows that there is some sort of carryover: the nitrate signal is larger during bromide ionization than it is when neither ion source is active

**Author reply:** When neither source is active the system measures only natural ambient ions, which results in many orders of magnitude lower signal level. The nitrate signal measured in bromide mode indicates that some nitric acid was present in the sampled gas stream and that the bromide ionized it. If nitric acid would be present at very large quantities, significant depletion of bromide ions would be expected, which was not observed.

Addition to manuscript text (page 4, line 168): When neither source is active, the system measures only natural ambient ions, which results in many orders of magnitude lower signal level. The rapid ion mode switch is completed within about a second timeframe, with minimal interference from the idle ion mode. The small nitrate ion signal measured during bromide mode indicates that some nitric acid was present in the sampled gas stream and is ionized by Br-.

**Other Comments:**

1. Line 242. This paragraph makes no sense: the authors claim that nitrate and bromide ionization have similar adduct binding strengths, and this somehow means that the two ionization techniques can be used. Why is this the case? Why can't quite different reagent ions be used in the system? Later in the paragraph, the authors seem to say that one can use reagent ions with different ionization mechanisms, so the authors need to clarify their point.

Author reply: Nitrate and bromide reagent ions do form adducts with similar binding strengths to the same product structures as indicated by the provided reference (Hyttinen et al. 2018). The text was meant to imply that as these products have similar binding strengths, they survive from similar energy collisions inside the mass spectrometer, and thus also fragment to a similar extent. As also the masses are very comparable (only separated by 17 Th) the transmission is likely comparable. Thus, there's a minimal detection bias between these ionization systems regardless of the mass spectrometer settings. If reagent ions with very different binding strengths would be used then most of the weaker binding product\*ion clusters would fragment with the same mass spectrometer settings, and thus a strong bias favoring the other technique would be recorded. Of course, it is possible to automatically change all the mass spectrometer settings before applying the second ion mode, but this would change also the transmission characteristics, leading to much more convoluted information due to many exchanging parameters between the two systems. However, these issues do not exclude usage of any combination of reagent ions – they are physical features that need to be considered in choosing a reagent ion combination that will perform best for the application at hand. If the ionization schemes work with different principles (e.g., first generates cluster ions whereas second results in fragment ions) then they are likely more readily applicable together as the detection is not hanging from a one parameter anymore (e.g., cluster binding strength).

To clarify the points raised up by the reviewer, additions were made into the manuscript text:

Addition to manuscript text (page 4, line 154): An ion combination which works well with the same mass spectrometer settings between the ionization schemes is highly beneficial, as in this way the ion transmission characteristics of the instrument are minimally affected, and the detected signal heights depend mainly on the individual product\*reagent ion binding strengths. This detection issue

can be largely avoided if the two ionization schemes work with different ionization mechanism, i.e., if the product detection sensitivity depends on the extent of fragmentation only in one mode and the other mode is barely influenced by the electric field strength change (e.g., if the first mode creates adducts and the second mode transfers charge, leading to ion-adducts and charged molecules, respectively). Moreover, the design of the MION is ideally suited for investigating the detailed influence of ion-molecule reactions and reaction times (and thus also ion-molecule reaction kinetics), enabled by using the same reagent ion precursor feed for multiple ion sources. While in principle any combination of reagent ions is possible in the MION, for this work Br- and NO3- were selected not only due to their differing ionization characteristics and good performance under identical mass spectrometer settings, but also for their potential to offer complementary insight into the inspected VOC oxidation processes.

2. The authors use calibrations of nitrate and bromide CIMS for sulfuric acid to prove that their instrument is capable of measuring this species. However, it is not clear if these experiments were done in the 'rapid switching' mode that is the core of the paper. In fact, the inset in Figure 3 suggests that the calibration was done independently for the two reagent ions. If this is the case, then this section does not support the central claim and focus of the manuscript that the switching reagent ion system provides quantitative measurements of sticky compounds like H2SO4! The measurements must be done in switching reagent mode. Please show the timeseries along with the calibration curves (i.e. I like the setup of Figure 3 - it just needs to demonstrate that these curves hold under the actual reagent ion switching setup, and at different relative humidities).

**Author reply:** The calibration measurements were done for each ion mode separately, as the purpose was to investigate the detection sensitivity and not the switching of the ion mode in this particular case. Due to the unavailable instrument time we are not able to provide a new calibration measurement with switching ion modes as requested by the reviewer. However, the same issue has already been demonstrated in Figure 5 in which the highly-oxidized, and thus arguably sticky, alphapinene derived peroxy radical  $C_{10}H_{17}O_7$  is detected in both ion modes, and thus switches between the spectra when ion mode is changed. Furthermore, the linearities of the calibration curves obtained for both ion modes (Figure 3) show that no apparent wall reaction related problems were encountered during the calibration measurement, notwithstanding the sticky nature of the photochemically generated sulfuric acid used in calibration.

3. Sulfuric acid detection by bromide reagent ions is a constructive direction - but needs more analysis to support the claims. Specifically, I would like to see: is there a dependence on relative humidity? And two what extent does the system form clusters vs bare ions?

**Author reply:** The measurement of sulfuric acid was carried out only to compare detection characteristics against a general standard, which has been used as a common yardstick to estimate HOM detection efficiency (see Ehn et al. 2014 and Bianchi et al. 2019). This practice is an idealization, with caveats that are known to the community. The measurement of sulfuric acid by atmospheric pressure negative polarity chemical ionization is likely possible with a vast number of different reagent ions, as the sulfuric acid molecule contains both two hydrogen bond donor and two acceptor sites, enabling it to cluster (=bind) efficiently with multiple different reagent ions.

It is common knowledge that humidity generally becomes a problem at higher levels with the APCI technique (see e.g. Iyer et al. 2018 and Hyttinen et al. 2018). Different reagents have different humidity dependences and finding the humidity responses for the two reagent ions selected for demonstrating the functionality of the inlet is not within the scope of this manuscript. Nevertheless, at this low humidity (RH max. about 1%) no dependence was observed or expected. With our mass spectrometer settings, sulfuric acid was apparently mainly observed as a cluster ion, and at the deprotonated mass a strong background signal from bromide-water cluster was hindering the retrieval.

4. Figure 4: I think the authors intend to show that peaks in the spectra are oxygenated organics as demonstrated by labels of O5 / O6 / O7 / etc. Are these peaks actually CxHyOz=5,6,7 or are they truly O5- ions? Please label according to actual chemical formulae.

**Author reply:** Yes, these are oxidized organics with the oxygen content of the product compound indicated. They certainly are not O5- ions or any other ionic oxygen clusters. A considerable effort has been devoted in previous publications to unravel their identity as covalently bound organic oxidation products and this analysis is not repeated here (see Bianchi et al. 2019 and references therein). We understand this could be a confusing labelling for anyone else not working with highly-oxidized reaction products such as HOM, and thus the notation has now been augmented. More details of the detected reaction products have been added to Figure 4 and its caption.

---

## Referee Report (RR1)

I wish to point out that I was not a reviewer of this manuscript in the first round of review. I find it a pleasure to read the updated manuscript. The novel inlet for fast switching of reagent ions is of great interest to the community. I only have minor comments.

1.      As the main focus of the manuscript is to introduce the inlet design, more details about the inlet should be provided, such as the dimensions of the source blocks. What's the weight of each ionization stage? This may be an issue when multiple ionization stages are deployed.

2.      From figure 1a, it looks like that the directions of two ionization stages are 90degree different. Is this for experimental reason or for illustration?

3.      Figure 2. It would be nice to include an insert figure with shorter time period, to better show the switch between two reagent ions.

4.      The authors hypothesized that the difference in sulfuric acid calibration factors between $Br^-$ and $NO_3^-$ is due to the ion-molecule reaction time. This is an easy hypothesis to test, just by switching the positions of two reagent ions.

---

## Author Response (AR2)

**Answers to the 2nd review comments on the manuscript:**

**Multi-scheme chemical ionization inlet (MION) for fast switching of reagent ion chemistry in atmospheric pressure chemical ionization mass spectrometry (CIMS) applications**

By: Matti P. Rissanen, Jyri Mikkilä, Siddharth Iyer and Jani Hakala

**Reviewer#1**

*The author has addressed my major comments and questions and the paper has greatly improved upon revision. I would like to recommend the manuscript for publication in AMT, but have one last suggestion:*
*Redo Figure S4 similarly as Figure 4 (mark the numbers of oxygen atoms and several major peaks, which would be essential information for the discussion).*

**Author reply:**

Figure S4 was redone in-line with the reviewer recommendation. Below are the old and new versions of the Figure S4 and their captions.

**New Figure S4 and its caption:**

[Figure]

**Figure S4** Example spectra obtained from cyclohexene ozonolysis experiments in both ion modes shown with a common product mass axis, *i.e.,* the Br- spectrum is displaced by 17 Th (=difference between reagent ion Br- and NO₃- masses) to overlap the same composition products horizontally. Upper panels show nitrate spectra (red) and lower panels bromide spectra (blue). *a)* Illustrates the reagent ion peaks, *b)* the monomer range (*i.e.,* oxidation products which have the same number or less carbon atoms than cyclohexene), and *c)* the dimer range (*i.e.,* oxidation products with about two times the carbon number of cyclohexene), respectively. For a few of the most prominent product peaks also the explicit compositions are shown.

**Old Figure S4 and its caption:**

[Figure]

**Figure S4** Example spectra obtained from cyclohexene ozonolysis experiments in both ion modes shown with a common product mass axis, *i.e.,* the Br- spectrum is displaced by 17 Th (=difference between reagent ion Br- and NO₃- masses) to overlap the same composition products horizontally. Upper panels show nitrate spectra (red) and lower panels bromide spectra (blue). *a)* Illustrates the reagent ion peaks, *b)* the monomer range (*i.e.,* oxidation products which have the same number or less carbon atoms than cyclohexene), and *c)* the dimer range (*i.e.,* oxidation products with about two times the carbon number of cyclohexene), respectively.

**Reviewer#2**

*Review of Rissanen et al.*

*I wish to point out that I was not a reviewer of this manuscript in the first round of review.*

*I find it a pleasure to read the updated manuscript. The novel inlet for fast switching of reagent ions is of great interest to the community. I only have minor comments.*

1. *As the main focus of the manuscript is to introduce the inlet design, more details about the inlet should be provided, such as the dimensions of the source blocks. What's the weight of each ionization stage? This may be an issue when multiple ionization stages are deployed.*

**Author reply:** More physical details were added to the main text, based on the reviewer recommendation. In addition, an additional schematic figure of the inlet with marked dimensions was added to the supplementary material.

**Addition to text (page 2, line 103):** "The ion source connecting MION to the mass spectrometer entrance weights about 2.1 kg, with every additional stage weighting roughly 1.5 kg each. The distance of the first ion source to the mass spectrometer inlet orifice is approximately 3 cm, and the length of an additional ionization stage is about 10 cm, with a height of 16 cm. A schematic Figure S1 with marked dimensions can be found in the supplementary material."

**New figure S1:**

[Figure]

**Figure S1** Schematic of the MION showing the approximate dimensions of the inlet design.

> 2. *From figure 1a, it looks like that the directions of two ionization stages are 90degree different. Is this for experimental reason or for illustration?*

**Author reply:** This is the schematic of the current design. The ion sources are 90degrees to each other to enable easier usage of the inlet connections. This should not matter to the measurement to any significant extent, as the flow inside the pipe has a circular symmetry.

> 3. *Figure 2. It would be nice to include an insert figure with shorter time period, to better show the switch between two reagent ions.*

**Author reply:** According to the reviewer recommendation we have added an insert figure to better demonstrate the rapid ion mode switch.

**New figure 2 and its caption:**

[Figure]

**Figure 2** An example of switching between multiple ion chemistries while sampling laboratory air. The ion modes utilized have been marked with separate colours and are further labelled above the figure with abbreviations: APi for ambient ion mode (no active ionization applied; black trace), Br- for bromide ion ionization mode (red trace), $NO_3$- for nitrate ion ionization mode (blue trace), and TIC is used to indicate total ion count measured (black trace). The insert demonstrates the rapid ion mode switch, shown in a linear scale. All signals shown here were retrieved by high resolution peak fitting.

**Old Figure 2 and its caption:**

[Figure]

**Figure 2** An example of switching between multiple ion chemistries while sampling laboratory air. The ion modes utilized have been marked with separate colours and are further labelled above the figure with abbreviations: APi for ambient ion mode (no active ionization applied; black trace), Br- for bromide ion ionization mode (red trace), $NO_3$- for nitrate ion ionization mode (blue trace), and TIC is used to indicate total ion count measured (black trace). All signals shown here were retrieved by high resolution peak fitting.

*4. The authors hypothesized that the difference in sulfuric acid calibration factors between Br- and NO3- is due to the ion-molecule reaction time. This is an easy hypothesis to test, just by switching the positions of two reagent ions.*

**Author reply:** We do agree with the reviewer that this would be a relatively easy task to do but unfortunately, we do not have the needed instrumentation to repeat the calibration measurement in the near future. We do appreciate the comment and will return to this issue when the needed equipment is again available to us.